

# Optimization of cultural conditions for pectinase production by *Diaporthe* isolate Z1-1N and its pathogenicity on kiwifruit

Shu-Dong Zhang, Ling-Ling Chen, Chao-Yue Li, Xiao-Qing Long, Xue Yang, Xiao-Duo He, Li-Wen Du, Heng-Feng Yang and Li-Zhen Ling

Key Laboratory for Specialty Agricultural Germplasm Resources Development and Utilization of Guizhou Province, Liupanshui Normal University, Liupanshui, Guizhou, China

## ABSTRACT

*Diaporthe* Z1-1N, the primary causal agent of soft rot disease in kiwifruit, exhibited higher pectinase activity compared to cellulase activity in both *in vitro* and *in vivo* incubation models. To gain deeper insights into the role of pectinases in the pathogenicity of this fungus, we evaluated the effects of incubation temperature (ranging from 18 to 38 °C), duration (1 to 7 days), and medium pH (4.0 to 9.0) on the activities of two crucial pectinases: polygalacturonase (PG) and polymethylgalacturonase (PMG). Our single-factor experiments revealed that the optimal conditions for maximizing PMG yield were a pH of 7.5 and a temperature of 28 °C, with peak activity occurring after three days of incubation. Notably, PG activity peaked on the fourth day under the same pH and temperature conditions. Under the optimal conditions identified through an orthogonal experimental design, PMG exhibited higher activity than PG. Further analysis showed that temperature was the most influential factor on PMG activity, followed by incubation duration and pH. The lesion size caused by the purified pectinase extracts was 50% the lesion size that caused by the fungal mycelium of *Diaporthe* Z1-1N. These findings underscore the significance of PG and PMG as key virulence factors in the pathogenicity of *Diaporthe* Z1-1N, providing a solid scientific basis for future research into the functions of these enzymes.

## INTRODUCTION

Kiwifruit (*Actinidia* spp.), commonly referred to as "mihoutao" (monkey peach) in China, represents an important genus within the family Actinidiaceae. The most recent taxonomic revision identifies 55 species and 20 varieties within this genus (*Li, Li & Soejarto, 2007*). Among these, *A. chinensis* and *A. deliciosa* are the predominant species cultivated, collectively accounting for nearly all kiwifruit in international trade (*Nishiyama, 2007*). Currently, *A. arguta* is cultivated on a smaller scale in Europe, New Zealand, and the United States, both commercially and by enthusiastic amateurs (*Ferguson, 2013*). Kiwifruit is renowned for its excellent flavor and is rich in vitamin C, minerals, dietary fiber, phenols, carotenoids, and other essential nutrients (*Liu et al., 2019*; *Ma et al., 2017*; *Nishiyama et*

Corresponding author
Li-Zhen Ling, 121302168@qq.com

*al., 2004*; *Sivakumaran et al., 2018*). However, soft rot disease poses a significant threat to the quality and yield of kiwifruit (*Li et al., 2016*). Research indicates that fungi from the *Diaporthe* spp. are among the primary agents responsible for this soft rot disease (*Díaz et al., 2014*; *Lei et al., 2019*; *Zhou et al., 2015*), resulting in considerable annual economic losses.

The genus *Diaporthe* comprises over 1,300 taxa, with *Phomopsis*, its asexual states, also encompassing more than 1,000 species listed in MycoBank (http://www.mycobank.org, accessed in December 2024), which exhibit broad host ranges and a global distribution. According to the International Code of Nomenclature for algae, fungi, and plants (*Turland et al., 2018*), *Diaporthe* takes precedence for recommendations regarding generic names due to the existence of two or more genera typified by either a sexual or asexual morph (*Rossman et al., 2015*). The *Diaporthe* genus encompasses a diverse group of fungi that can exist in various forms, including as endophytes, saprotrophs, and plant pathogens. Research has shown that some of the most harmful fungal pathogens can be capable of causing various diseases, including stem canker, leaf spot blight, and fruit decay (*Bai et al., 2016*; *Li et al., 2017*; *Díaz et al., 2017*; *Wan et al., 2022*). Currently, more than 10 *Diaporthe* species have been reported to cause soft rot disease during both the growing season and post-harvest storage of kiwifruit (*Ling et al., 2024b*). These pathogens can secrete cell wall degrading enzymes (CWDEs) that interact with host cell wall constituents, thereby facilitating pathogen penetration through the loosened host cell walls and middle lamella matrices. The CWDEs synthesized during plant infection by *Diaporthe* spp. include cellulases, xylanases, and pectinases (*Chen et al., 2018*; *Zhang, 2010*).

Pectinases are pivotal in plant pathogenesis, marking the initial enzymes synthesized by specific fungal and bacterial pathogens that colonize plant cell walls, priming these cell wall components for subsequent degradation by other enzymes (*Abbott & Boraston, 2008*). Typically, pectin degradation is orchestrated by a suite of pectinases, which include pectate lyases (families 1, 2, 3, and 9), polygalacturonases and rhamnogalacturonases (glycoside hydrolase family 28), pectin methylesterases (carbohydrate esterase family 8), and pectin acetylesterases (carbohydrate esterase family 12) (CAZY Database; https://www.cazy.org/). These enzymes are hypothesized to contribute to soft rot infections by dismantling cell wall polysaccharides, leading to fruit softening and a decrease in disease resistance. Polygalacturonase (PG), pectinesterase (PE), and polymethylgalacturonase (PMG) are particularly noted for their roles in facilitating pectin disassembly in postharvest fresh fruits (*Lin et al., 2018*). Our recent research has shown that kiwifruit inoculated with *Diaporthe* Z1-1N displayed a higher disease index compared to non-inoculated harvested kiwifruit, with two pectinases (PG and PMG) showing increased activities in both *in vitro* and *in vivo* incubation models (*Chen et al., 2024*; *Ling et al., 2024a*). This finding indicates that these enzymes might have the significant role in the pathogenic process and warrant further study.

Previous studies have indicated that pectinase activity levels are influenced by factors such as $Ca^{2+}$, pH, and the culture medium (*Pagel & Heitefuss, 1990*; *Qi et al., 2010*). To better understand the potential role of the enzymes as virulence factors under various conditions in the fungi-host plant interaction, it is essential to optimate the culture

conditions of these two pectinase activity levels. Therefore, this study aims to examine the activities of two pectinases, PG and PMG, at the different incubation temperature, incubation time, and the initial pH of the culture medium through single-factor and orthogonal optimization experiments. Specifically, we investigated the effects of various inculcation factors on the production of PMG and PG. Additionally, the pathogenicity of pectinase produced by *Diaporthe* isolate Z1-1N under the optimal condition was assessed and compared with that of mycelial plugs from the *Diaporthe* isolate Z1-1N. This research will lay the foundation for further exploring the pathogenic role of these two pectinases in the soft rot of 'Hongyang' kiwifruit.

## MATERIALS AND METHODS

### Plant material and pathogenic fungi

Kiwifruit (*A. chinensis* cv. Hongyang) of similar size and consistent maturity was purchased from the local market in Liupanshui, Guizhou Province, China. The pathogenic fungi *Diaporthe* spp. Z1-1N, isolated in our previous studies from rotten 'Hongyang' kiwifruit during cold storage (*Ling et al., 2023*; *Ling et al., 2024b*), was cultured on potato dextrose agar (PDA) medium at 25 °C in the dark for 6 days for the enzyme assays and pathogenicity trials.

### Optimization of pectinase production in liquid culture

In this study, we employed a modified Marcus liquid medium, as described in our previous research, to investigate the production of two pectinases: PG and PMG (*Ling et al., 2024a*). Five millimeter mycelial discs from a 6-day-old PDA culture were inoculated into 100 ml of liquid medium contained in 250 ml flasks, which were then incubated on an orbital shaker. Optimization of pectinase production was conducted through a series of single factor experiments, assessing variables such as incubation periods (1, 2, 3, 4, 5, 6, and 7 days), temperature (18, 23, 28, 33, and 38 °C), and the initial pH of the medium (ranging from 4 to 9 in 0.5 intervals). The optimal incubation time was determined based on the experimental design. When investigating the other two factors, cultures were harvested on the third day. The contents were filtered using a vacuum to remove the fungal mycelia. The filtrates were subsequently centrifuged at 10,000 g for 15 min at 4 °C, and the supernatant was utilized to assess the activitiesof PMG and PG using methods described in our previous study (*Ling et al., 2024a*).

Based on the results of the single-factor experiment, a three-factor, three-level standard orthogonal table, $L_9$ $(3^3)$, was selected to optimize the production conditions for two pectinases (PG and PMG) as shown in Table 1. The incubation temperature (A) was set at 25 °C, 28 °C, and 31 °C; the initial pH of the medium (B) was set at 6.5, 7.0, and 7.5; and the incubation time was set at 2, 3, and 4 days. These factors and their corresponding levels were incorporated into the $L_9$ $(3^3)$ orthogonal table, resulting in a total of nine experimental groups with various combinations of the three parameters in this study.

**Table 1  Factors and levels of pectinase production.**

| Levels | Factors | | |
| --- | --- | --- | --- |
| | Inoculation temperature (°C) | Initial pH of the medium | Inoculation time (d) |
| | A | B | C |
| 1 | 25 | 6.5 | 2 |
| 2 | 28 | 7.0 | 3 |
| 3 | 31 | 7.5 | 4 |

## Measurement of pectinase activity

The activities of PMG and PG were determined using the 3,5-dinitrosalicylic acid (DNS) colorimetric method, following the detailed procedures outlined by our previous study (*Ling et al., 2024a*). Three biological trials were conducted.

## Purification of crude enzyme extracts and incubation on kiwifruit fruit

Pectinase was induced for production from the liquid culture of the *Diaporthe* isolate Z1-1N using orange pectin (Shanghai Aladdin Industries, Shanghai, China) as the carbon source. The purification process adhered to the method described (*Ling et al., 2024a*). The pathogenicity of various treatments, including pectinase extract and mycelial plugs from the *Diaporthe* isolate Z1-1N, was assessed by incubating them at 28 °C in darkness. Sterile water and sterile plugs served as controls, respectively. Ten kiwifruit fruits were used in each treatment. The lesions are classified based on their diameters (*Li et al., 2019*): grade 0 (0 cm); grade 1 (0 cm−1.5cm); grade 3 (1.5–3 cm); grade 5 (3–4.5 cm); grade 7 (4.5–6 cm); grade 9 (more than six cm). The Disease Index (DI) is calculated as follows:

$$\text{Disease Index (DI)} = \frac{\sum(\text{The rotten fruits of each grade} \times \text{the corresponding grade})}{\text{Total number of tested fruits} \times \text{the topmost grade}} \times 100\%.$$

## Data and statistical analysis

Each experiment of enzymic assay was conducted in triplicate, and lesion size measurements were taken from ten kiwifruit fruits per treatment group. From the gathered data, we calculated the mean and standard error to provide a comprehensive analysis. To discern significant differences among the means, we employed one-way analysis of variance (ANOVA) using SPSS version 19.0 for Windows. Following the ANOVA, we applied Duncan's multiple range test to separate the means, establishing statistical significance at the $P < 0.05$ threshold. Additionally, we leveraged the Orthogonal Designing Assistant II V3.1 software for the rigorous evaluation of our statistical experimental design, ensuring the precision of our experimental outcomes.

## RESULTS

### Effect of initial pH of medium on two pectinase production

The initial pH of the medium is a crucial factor in the production of pectinases, as it influences both the type and quantity of enzymes produced by fungi. In this study, we
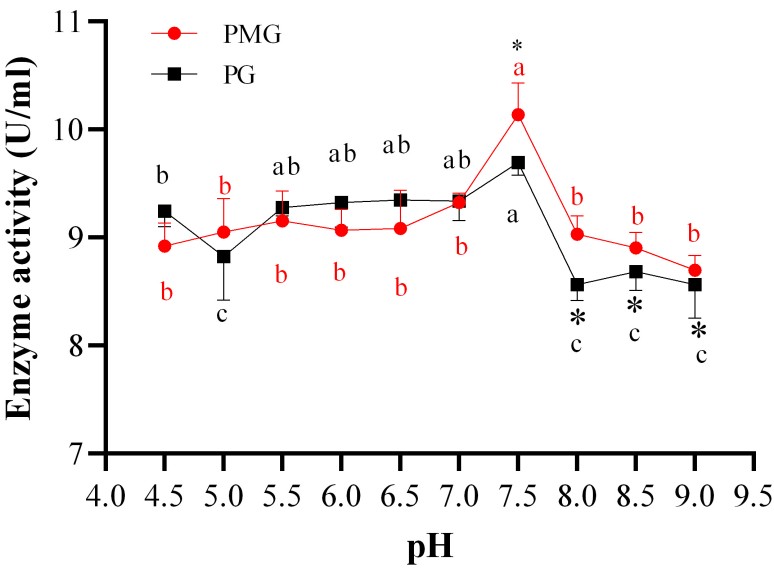

**Figure 1 Effect of pH on PMG and PG production.** Note: Different letters in each line indicate significant difference of each enzyme ($P < 0.05$); an asterisk (*) indicates the significant difference of two enzymes ($P < 0.05$).

found that the activities of pectin methylgalacturonase (PMG) and polygalacturonase (PG) produced by *Diaporthe* Z1-1N showed the similar trends but also exhibited some differences (Fig. 1). For example, significant differences in enzyme activities were only observed at pH levels above 7.5, specifically in the range of pH 7.5 to 8.0. Within this range (7.5–9.0), PG activity significantly decreased compared to that of PMG (Fig. 1). In addition, PMG activity remained relatively stable across different pH levels, with a significant peak at pH 7.5 (10.14 U/ml). Similarly, PG activity reached its highest level at pH 7.5 (9.70 U/ml) but it was lower than that of PMG. Therefore, these results indicate that the optimal production of both PMG and PG by *Diaporthe* Z1-1N occurs at pH 7.5, with PMG exhibiting a higher activity than PG.

### Effect of incubation temperature on two pectinase production

Temperature is directly related to the metabolic activities of microorganisms and significantly influences both growth and product formation. Figure 2 illustrates the effects of various temperatures on the production of two types of pectinase. The activity of PMG was consistently higher than that of PG across different temperatures, with the exception of 33 °C, where PG exhibited greater activity than PMG. Additionally, both PMG and PG displayed a similar trend, characterized by an initial increase in activity followed by a decline. Our results indicate that maximum pectinase production occurred at 28 °C, yielding 10.14 U/ml for PMG and 9.70 U/ml for PG. Subsequently, the second most favorable temperature for PMG production was 23 °C (8.87 U/ml), while for PG, it was 33 °C (8.92 U/ml). Conversely, the lowest production levels for both pectinases were observed at 38 °C, with values of 7.32 U/ml for PMG and 6.39 U/ml for PG. Therefore,
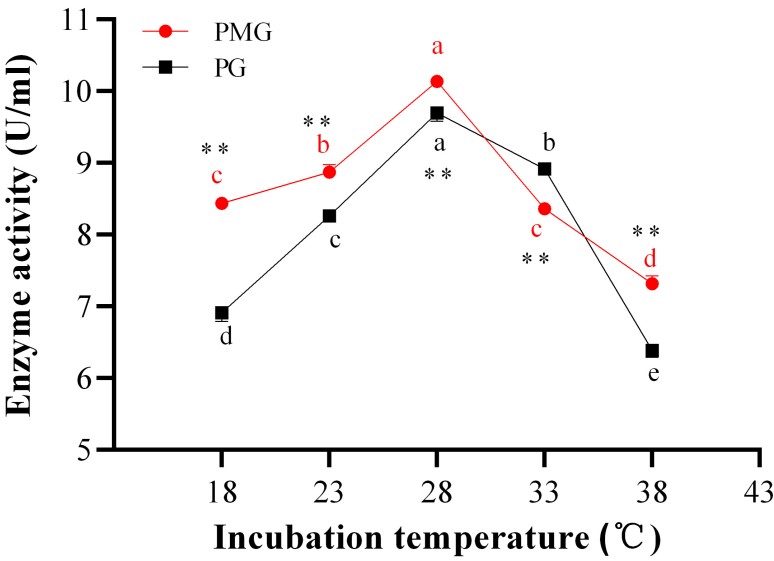

**Figure 2** **Effect of incubation temperature on PMG and PG production.** Note: Different letters in each line indicate significant difference of each enzyme ($P < 0.05$); two asterisks (**) indicate the significant difference of two enzymes ($P < 0.01$).

these findings suggest that the optimal temperature for the growth and production of PMG and PG by *Diaporthe* Z1-1N is 28 °C.

## Effect of incubation periods on two pectinase production

The incubation period is crucial for maximizing enzyme yield. Over the course of 1 to 7 days, we monitored the activities of PMG and PG, as illustrated in Fig. 3. Throughout this period, PMG generally exhibited a higher activity than PG, with the exception of the second and fifth days. Our data further indicated that both pectinases displayed significant fluctuations in activity and showed a similar trend with a slight variation. For PG, activity gradually increased from the start, peaking on the third day at a concentration of 10.52 U/ml, before gradually declining by the seventh day. PMG activity began to rise on the second day, dipped slightly on the third day, and then reached its maximum on the fourth day at 10.43 U/ml. Subsequently, PMG activity decreased, reaching its lowest point at the 7-day mark with a yield of 7.42 U/ml, as depicted in Fig. 3. These findings suggest that the optimal times for PMG and PG production are the third and fourth days of incubation, respectively.

## Optimization of cultural conditions for pectinase productivity

Culture conditions, such as pH levels, incubation temperatures, incubation duration, and the sources of carbon, nitrogen, and mineral salts, are pivotal in dictating the synthesis and secretion of extracellular enzymes by microorganisms (*Jia et al., 2010*; *Koirala et al., 2014*; *Patidar et al., 2018*). Hence, meticulous adjustment of these parameters can obtain the highest activity the major extracellular enzymes secreted by pathogenic fungi. An

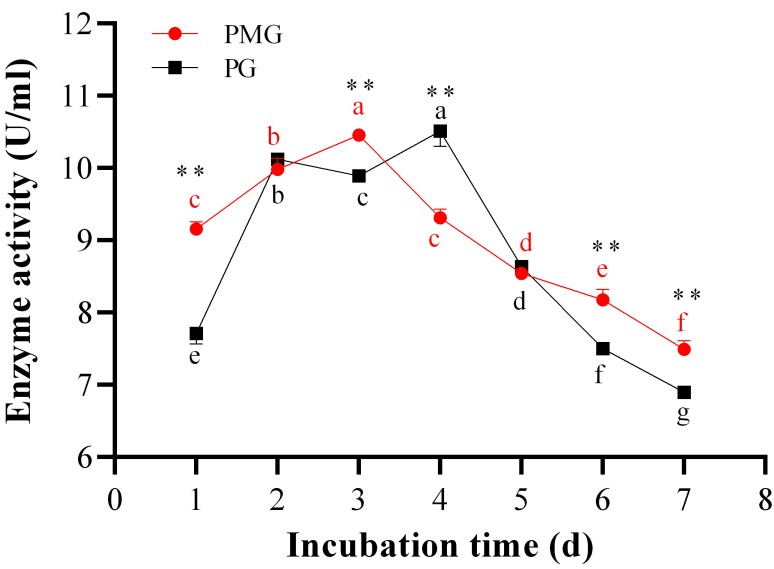

**Figure 3** **Effect of incubation time on PMG and PG production.** Note: Different letters in each line indicate significant difference of each enzyme ($P < 0.05$); two asterisks (**) indicate the significant difference of two enzymes ($P < 0.01$).

**Table 2** **Results of the orthogonal experimental design.**

| Tested no. | Factors | | | Enzymatic activity (U/mL) | |
|---|---|---|---|---|---|
| | **A** | **B** | **C** | **PMG** | **PG** |
| 1 | 25 | 6.5 | 2 | 9.702 | 9.027 |
| 2 | 25 | 7.0 | 3 | 9.755 | 9.213 |
| 3 | 25 | 7.5 | 4 | 9.306 | 7.864 |
| 4 | 28 | 6.5 | 3 | 10.949 | 9.999 |
| 5 | 28 | 7.0 | 4 | 10.002 | 9.415 |
| 6 | 28 | 7.5 | 2 | 11.228 | 10.056 |
| 7 | 31 | 6.5 | 4 | 8.730 | 6.141 |
| 8 | 31 | 7.0 | 2 | 10.661 | 8.536 |
| 9 | 31 | 7.5 | 3 | 9.467 | 8.384 |

**Notes.**

Factor A: Inoculation temperature (°C); Factor B: Initial pH of the medium; factor C: Inoculation time (d).

orthogonal experiment demonstrated that the highest activity of PMG, at 11.228 U/mL, was attained at a temperature of 28 °C, an initial pH of 7.5, and an incubation period of 2 days (Table 2). Under these same conditions, PG achieved its peak activity of 10.056 U/mL (Table 2). This finding demonstrated that under the optimal conditions identified through an orthogonal experimental design, PMG exhibited a higher activity than PG.

In this study, we examined the impact of three key factors—incubation temperature (A), initial medium pH (B), and incubation duration (C)—on PMG production by *Diaporthe* Z1-1N. A higher extreme *R* value for a factor indicates that it has a more significant effect on the outcome. By analyzing the magnitude of the extreme R values, we ranked the influence

**Table 3 Analysis of orthogonal experiment of PMG activity.**

| | PMG activity (U/mL) | | |
|---|---|---|---|
| | **A** | **B** | **C** |
| K1 | 9.59 | 9.79 | 10.53 |
| K2 | 10.73 | 10.14 | 10.06 |
| K3 | 9.62 | 10.00 | 9.35 |
| Range (R) | 1.14 | 0.35 | 0.58 |
| The factor importance | | A>C>B | |
| Optimal combination | | $A_2B_2C_1$ (11.50 U/mL) | |

**Notes.**
Factor A: Inoculation temperature (°C); Factor B: Initial pH of the medium; factor C: Inoculation time (d).

**Table 4 Disease degree of 'Hongyang' kiwifruit under different treatments.**

| Treatments | DI (%) |
|---|---|
| Sterile water | 0.00 |
| Pectinase extract | 47.62 |
| Sterile plug | 0.00 |
| Mycelial plug | 94.81 |

**Notes.**
The data was obtained after seven days of the different treatment.

of these factors on PMG production activity as follows: A >C >B (Table 3). This ranking indicates that incubation temperature had the greatest impact, followed by incubation duration, and then initial pH value. As previously mentioned, the optimal conditions were determined to be an incubation time of 2 days, an incubation temperature of 28 °C, and an initial pH of 7.0. To confirm the effectiveness of the optimization approach based on the orthogonal experiment's results, we conducted a verification experiment using the refined parameter combination. The findings revealed that PMG activity increased significantly to 11.50 U/mL (Table 3), surpassing the activity observed in all previous orthogonal experiments. This outcome validates the efficacy of the optimization strategy employed.

## Pathogenicity of the pectinase extract on kiwifruit

In this study, we investigated the effects of a purified pectinase extract on kiwifruit. The pectinase was prepared from a three-day shaken lipid culture using orange pectin as the sole carbon source, based on the optimal conditions for PMG production. Our results indicated that the pectinase extract induced significant necrotic lesions, with the DI value reaching 47.62% after seven days of treatment (Table 4). Additionally, we observed that the symptoms caused by the pectinase extract were similar to those resulting from infection with *Diaporthe* Z1-1N mycelium. However, the necrotic lesions induced by the pectinase extract were considerably less severe than those caused by the mycelium plug, which exhibited a DI of 94.81%. In contrast, the control fruits inoculated with sterile water or PDA plugs showed no lesions. Therefore, these results suggest that pectinase plays a role in lesion development associated with *Diaporthe* Z1-1N.

## DISCUSSION

One of the primary barriers against phytopathogenic fungi is the plant cell wall, which is rich in polysaccharides. Most fungi must breach this barrier to access plant cells, necessitating the secretion of various enzymes capable of degrading cell wall polymers. Pectinases, a group of enzymes that decompose pectic substances, are classified into several categories: polygalacturonase (PG), polymethylgalacturonase (PMG), pectin lyase (PL), pectate lyase (PAL), and pectin methylesterase (PME) (*Xue et al., 2018*). Our recent study reported that the activities PG and PMG produced by *Diaporthe* Z1-1N exhibited the significant increase from 3 to 5 days when this fungi was cultivated in a kiwifruit infection model (*Ling et al., 2024a*). Furthermore, the production of these enzymes secreted by this fungus was biochemically assayed in an *in vitro* system (*Ling et al., 2024a*). To determine the optimal external conditions for enzyme production, this study aims to investigate the production of these two pectinases by *Diaporthe* Z1-1N under varying incubation temperatures (18–38 °C), incubation durations (1–7 days), and pH levels of the medium (4.0–9.0).

Our findings indicated that during the single-factor experiment, both PMG and PG achieved their maximum activities at the same incubation temperature of 28 °C and medium pH of 7.5. However, there was a slight variation in the timing of these peak activities over the incubation period. Specifically, PMG reached its peak on the third day, whereas PG attained its maximum on the fourth day. Previous research has established a sequence for the appearance of cell wall-degrading enzymes in phytopathogenic fungi (*Lisker et al., 1975*; *Lisker, Katan & Henis, 1975*). PG is widely recognized as a key virulence factor and is known to be induced in the early stages of infection (*Lisker et al., 1975*; *Vyas, Shah & Jayawant, 2025*). Indeed, our recent study has shown thatPMG and PG enzymes exhibited higher accumulation during the 3- to 5-day period of the infection of *Diaporthe* Z1-1N, which is associated with the development of kiwifruit soft rot. Furthermore, the levels of cell wall-degrading enzyme (CWDE) activity have been correlated with the severity of pathogenesis (*Gawade et al., 2017*; *Zhang, Bruton & Biles, 2014*; *Zhou et al., 2016*). Numerous studies have reported that PG activity is essential for full virulence across a range of host plants (*Kubicek, Starr & Glass, 2014*; *Vyas, Shah & Jayawant, 2025*). In this study, under the same incubation temperature and medium pH, PMG exhibited higher activities compared to PG. Similarly, during the infection of *Diaporthe* Z1-1N, PMG exhibited the larger activity levels than PG (*Ling et al., 2024a*). Therefore, we inferred that PMG enzyme is likely a significant virulence factor that should not be overlooked. The optimal conditions for PMG production by *Diaporthe* Z1-1N, as determined through an orthogonal experimental design, significantly enhanced its activity in an *in vitro* cultivated model. Additionally, the purified pectinase extracts resulted in a 50% lesion size of fungal mycelium. Both PG and PMG are crucial virulence determinants were observed in *Rhizoctonia solani* Kühn (*Chen et al., 2006*). Whether these two enzymes as crucial virulence determinants in the pathogenicity of *Diaporthe* Z1-1N requires further investigation. However, other pectinases, such as PME, PL, and $\beta$-galactosidase ($\beta$-gal), are also involved in disease development caused by phytopathogenic fungi (*Gawade et al.,*

*2017*). Consequently, further investigation into the roles of these additional enzymes is warranted.

Moreover, varying cultural conditions have led to differences in the activities and types of cell wall-degrading enzymes produced. For instance, PMG exhibited a higher activity when soluble starch was used as a carbon source, while PG demonstrated a greater activity when utilizing pectin as a substrate during *in vitro* cultivation of *Phomopsis longanae* Chi (*Chen et al., 2018*). Additionally, some other cultivation factors, such as metal ion concentrations ($Ca^{2+}$, $Fe^{2+}$) and shaking conditions, also influence the production of pectinases (*Chen et al., 2006*; *Pagel & Heitefuss, 1990*). Consequently, further investigation is needed to elucidate the relationship between specific cultural conditions and the virulence of cell wall-degrading enzymes. Notably, our result of orthogonal experiments indicated incubation temperature was the most influential factor for PMG production by *Diaporthe* Z1-1N among the three tested factors. Previous research has indicated that temperature affects the mycelial growth of *Sclerotium rolfsii* and the severity of rot on potato tubers (*Daami-Remadi et al., 2010*). Given that *Diaporthe* Z1-1N was isolated from rotten kiwifruit during cold storage, it is essential to investigate the effects of low temperature on the activities of PG and PMG. Additionally, pathogenicity tests should be conducted to better understand the role of these enzymes under such conditions.

## CONCLUSIONS

In this study, we optimized the production of two pectinase enzymes, PG and PMG, from *Diaporthe* Z1-1N, the microorganism responsible for soft rot disease in kiwifruit. This was achieved by evaluating three key factors—pH, temperature, and incubation duration—using an orthogonal experimental design. Our findings revealed that the highest production of PMG occurred at pH 7.0 and 28 °C on the third day, whereas maximum PG production was observed at the same pH and temperature but on the fourth day. Notably, PMG demonstrated the higher activity compared to PG when each factor was optimized individually. A more detailed analysis of PMG activity, based on the orthogonal design, indicated that incubation temperature was the most influential factor, followed by incubation time and pH. The purified pectinase produced by *Diaporthe* Z1-1N under the optimized culture conditions exhibited significant pathogenicity, as demonstrated by a 50% lesion size of the fungal mycelium of this fungi. These results provide a solid foundation for future research into the functions of PMG and PG.

### Funding

This work was supported by the Guizhou Science and Technology Department, grant number QianKeHeJiChu-ZK[2022]530, the Scientific Research (Cultivation) Project of Liupanshui Normal University, grant number LPSSY2023KJZDPY06 and College Student Innovation and Entrepreneurship Project (S202310977123, S202310977124, 2024109770677, 2024109770686, 2024109770692 and S2024109771629). The funders had

no role in study design, data collection and analysis, decision to publish, or preparation of the manuscript.

## Grant Disclosures
The following grant information was disclosed by the authors:
Guizhou Science and Technology Department: QianKeHeJiChu-ZK[2022]530.
Scientific Research (Cultivation) Project of Liupanshui Normal University: LPSSY2023KJZDPY06.
College Student Innovation and Entrepreneurship Project: S202310977123, S202310977124, 2024109770677, 2024109770686, 2024109770692, S2024109771629.

## Competing Interests
The authors declare there are no competing interests.

## Author Contributions

- Shu-Dong Zhang analyzed the data, authored or reviewed drafts of the article, and approved the final draft.
- Ling-Ling Chen analyzed the data, prepared figures and/or tables, and approved the final draft.
- Chao-Yue Li performed the experiments, prepared figures and/or tables, and approved the final draft.
- Xiao-Qing Long performed the experiments, prepared figures and/or tables, and approved the final draft.
- Xue Yang performed the experiments, prepared figures and/or tables, and approved the final draft.
- Xiao-Duo He performed the experiments, prepared figures and/or tables, and approved the final draft.
- Li-Wen Du performed the experiments, prepared figures and/or tables, and approved the final draft.
- Heng-Feng Yang performed the experiments, prepared figures and/or tables, and approved the final draft.
- Li-Zhen Ling conceived and designed the experiments, authored or reviewed drafts of the article, and approved the final draft.

## Data Availability
The raw measurements are available in the Supplementary File.

## Supplemental Information
Supplemental information for this article can be found online at http://dx.doi.org/10.7717/peerj.19207#supplemental-information.

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
