# Peer review of "Optimization of cultural conditions for pectinase production by Diaporthe isolate Z1-1N and its pathogenicity on kiwifruit"

_PeerJ, doi:10.7717/peerj.19207_

## Round 0.1 · original submission · Minor Revisions

Two reviewers evaluated your manuscript and found merit in the content. There are several concerns related to manuscript organization, lack of methodological details, and statements about biological replications, among others. One relevant aspect is the high self-citation rate, which does not allow the placement of this research in a broader context.

·

Basic reporting

I have made specific comments on the pdf attached.
• This manuscript examines the production of two enzymes by a Diaporthe isolate. Production in liquid medium was assessed under varying temperature, pH and time.
• The manuscript is generally quite well written and understandable. In a few places the narrative is not especially logical, especially where the authors try and explain the rationale for doing the work. See the end of the Introduction for example.
• The relationship between the work presented here and that which has come before (from the authors) is not always perfectly clear. For example, how the Diaporthe isolate was isolated and identified. How were the enzyme assays performed? How many reps?
• The work feels somewhat inward looking and self-referencing. In places it would benefit from better linkage to prior work. For example, I didn’t get a great understanding of prior work with such enzymes in other fungi.
• The work is technically straightforward. The techniques used are very simple and the results themselves are – for better or worse - rather samey and unremarkable. Overall, unfortunately, I think this is a very small piece of work and is really insufficient for a publication. This reads like an exploratory work before something further is performed.
• In places the outcomes of the enzyme assays are incorrectly linked directly to pathogenicity (see comments on doc).

Experimental design

above

Validity of the findings

above

Additional comments

above

·

Basic reporting

1. The manuscript is well-structured and written in clear and professional English. The introduction provides a strong background, contextualizing the importance of Diaporthe species in kiwifruit diseases and the role of pectinases in plant pathogenicity. Relevant literature is cited, including previous studies on Diaporthe infections and enzymatic degradation of plant cell walls, supporting the study's rationale.

2. While the introduction effectively summarizes past research, a clearer distinction of how this study builds upon or differs from previous findings would strengthen the presentation.

3. The paper follows a standard format, with clearly labeled figures and tables that appropriately illustrate the results. The inclusion of an orthogonal experimental design to optimize cultural conditions for enzyme production is particularly well-supported by the data presented.

4. While the manuscript mentions raw data availability, ensuring that all underlying data is fully accessible in compliance with the journal’s policies would be beneficial.

5. There are a few instances of awkward phrasing and grammatical inconsistencies that could be revised for better readability, a few examples:

- In the Abstract, the sentence "The purified pectinase extracts exhibited an impact of up to 50% on the lesion size of the fungal mycelium of Diaporthe Z1-1N." This phrasing is unclear; "exhibited an impact of up to 50%" is vague and could be reworded to: "The purified pectinase extracts increased lesion size in kiwifruit by up to 50%."
- In the Results section: "Under the optimal conditions identified through an orthogonal experimental design, PMG exhibited higher activity than PG." This sentence could be clearer by specifying how much higher the PMG activity was compared to PG.
- In the Discussion, the phrase: "These findings emphasize that both PG and PMG are crucial virulence determinants in the pathogenicity of Diaporthe Z1-1N, aligning with observations made in in Rhizoctonia solani Kühn." contains a repetition error ("in in"), which should be corrected to: "These findings emphasize that both PG and PMG are crucial virulence determinants in the pathogenicity of Diaporthe Z1-1N, aligning with observations made in Rhizoctonia solani Kühn."

A thorough proofreading pass would help improve clarity and readability.

Experimental design

1. The study presents original primary research that falls within the scope of the journal. The research question is well-defined, investigating the impact of temperature, pH, and incubation time on the production of two key pectinases, polygalacturonase (PG) and polymethylgalacturonase (PMG), and assessing their role in Diaporthe pathogenicity.

2. The methodology is rigorous, incorporating single-factor experiments and an orthogonal experimental design to determine optimal conditions for enzyme production. The study is strengthened by the inclusion of a pathogenicity assay, which evaluates the effect of purified pectinase extracts on lesion formation in kiwifruit.

3. The methods are described in sufficient detail to allow replication, specifying culture conditions, enzyme assays, and statistical analyses. The use of ANOVA and Duncan’s multiple range test is appropriate for assessing significant differences, and the study’s experimental design ensures control over key variables.

4. However, while the sample sizes for experiments are included in the raw data, they are not clearly stated in the main text. Making this information more explicit in the Methods section would improve clarity and accessibility for readers who may not immediately refer to the supplementary data. A brief addition such as: "Each experiment was conducted in triplicate, and lesion size measurements were taken from ten kiwifruit per treatment group," would provide essential context without requiring the reader to consult external files.

Validity of the findings

1. The results are well-supported by the data, demonstrating that temperature has the most significant effect on pectinase activity, followed by incubation duration and pH. The findings align with existing research on fungal cell wall-degrading enzymes, reinforcing the role of PG and PMG as virulence factors in Diaporthe species.

2. The pathogenicity assay provides additional evidence that pectinases contribute to lesion development in kiwifruit, with the purified enzyme extract increasing lesion size by up to 50%. However, the severity of damage caused by the enzyme extract is lower than that induced by the fungal mycelium, suggesting that additional virulence factors are involved in the infection process.

3. The statistical analysis is sound, and the data is presented in a logical and organized manner.

4. The conclusions accurately reflect the study’s findings and remain within the scope of the research question, avoiding overstatements regarding novelty or impact. However, expanding the discussion to explore the broader implications of these findings, particularly in relation to disease management strategies or the potential role of other enzymatic factors, would provide greater depth.

5. For example, discussing whether inhibitors targeting pectinase activity could be used as a control method for Diaporthe infections in kiwifruit production would enhance the practical relevance of the research.

Additional comments

1. This study is well-executed, with a clear research focus, a strong experimental design, and meaningful results.

2. The manuscript effectively presents its findings without exaggeration, and its conclusions are well-supported by the data.
3. Minor improvements in language clarity, raw data accessibility, and discussion expansion would enhance the paper further.

4. Making sample sizes more evident in the Methods section, refining some awkward phrasing, and expanding the discussion on practical applications would help strengthen the overall impact of the manuscript.

---

## Round 0.2 · accepted · Accept

The authors modified the manuscript following the Reviewers' suggestions. Consequently, the manuscript is ready for the next editorial stage.